# Learning from Limited Demonstrations

**Beomjoon Kim**
School of Computer Science
McGill University
Montreal, Quebec, Canada

**Amir-massoud Farahmand**
School of Computer Science
McGill University
Montreal, Quebec, Canada

**Joelle Pineau**
School of Computer Science
McGill University
Montreal, Quebec, Canada

**Doina Precup**
School of Computer Science
McGill University
Montreal, Quebec, Canada

## Abstract

We propose a Learning from Demonstration (LfD) algorithm which leverages expert data, even if they are very few or inaccurate. We achieve this by using both expert data, as well as reinforcement signals gathered through trial-and-error interactions with the environment. The key idea of our approach, Approximate Policy Iteration with Demonstration (APID), is that expert's suggestions are used to define linear constraints which guide the optimization performed by Approximate Policy Iteration. We prove an upper bound on the Bellman error of the estimate computed by APID at each iteration. Moreover, we show empirically that APID outperforms pure Approximate Policy Iteration, a state-of-the-art LfD algorithm, and supervised learning in a variety of scenarios, including when very few and/or suboptimal demonstrations are available. Our experiments include simulations as well as a real robot path-finding task.

## 1   Introduction

Learning from Demonstration (LfD) is a practical framework for learning complex behaviour policies from demonstration trajectories produced by an expert. In most conventional approaches to LfD, the agent observes mappings between states and actions in the expert trajectories, and uses supervised learning to estimate a function that can approximately reproduce this mapping. Ideally, the function (i.e., policy) should also generalize well to regions of the state space that are not observed in the demonstration data. Many of the recent methods focus on incrementally querying the expert in appropriate regions of the state space to improve the learned policy, or to reduce uncertainty [1, 2, 3]. Key assumptions of most these works are that (1) the expert exhibits optimal behaviour, (2) the expert demonstrations are abundant, and (3) the expert stays with the learning agent throughout the training. In practice, these assumptions significantly reduce the applicability of LfD.

We present a framework that leverages insights and techniques from the reinforcement learning (RL) literature to overcome these limitations of the conventional LfD methods. RL is a general framework for learning optimal policies from trial-and-error interactions with the environment [4, 5]. The conventional RL approaches alone, however, might have difficulties in achieving a good performance from relatively little data. Moreover, they are not particularly cautious to risk involved in trial-and-error learning, which could lead to catastrophic failures. A combination of both expert and interaction data (i.e., mixing LfD and RL), however, offers a tantalizing way to effectively address challenging real-world policy learning problems under realistic assumptions.

Our primary contribution is a new algorithmic framework that integrates LfD, tackled using a large margin classifier, with a regularized Approximate Policy Iteration (API) method. The method is

formulated as a coupled constraint convex optimization, in which expert demonstrations define a set of linear constraints in API. The optimization is formulated in a way that permits mistakes in the demonstrations provided by the expert, and also accommodates variable availability of demonstrations (i.e., just an initial batch or continued demonstrations). We provide a theoretical analysis describing an upper bound on the Bellman error achievable by our approach.

We evaluate our algorithm in a simulated environment under various scenarios, such as varying the quality and quantity of expert demonstrations. In all cases, we compare our algorithm with Least-Square Policy Iteration (LSPI) [6], a popular API method, as well as with a state-of-the-art LfD method, Dataset Aggregation (DAgger) [1]. We also evaluate the algorithm's practicality in a real robot path finding task, where there are few demonstrations, and exploration data is expensive due to limited time. In all of the experiments, our method outperformed LSPI, using fewer exploration data and exhibiting significantly less variance. Our method also significantly outperformed Dataset Aggregation (DAgger), a state-of-art LfD algorithm, in cases where the expert demonstrations are few or suboptimal.

## 2  Proposed Algorithm

We consider a *continuous-state, finite-action discounted MDP* $(\mathcal{X}, \mathcal{A}, P, \mathcal{R}, \gamma)$, where $\mathcal{X}$ is a measurable state space, $\mathcal{A}$ is a finite set of actions, $P : \mathcal{X} \times \mathcal{A} \rightarrow \mathcal{M}(\mathcal{X})$ is the transition model, $\mathcal{R} : \mathcal{X} \times \mathcal{A} \rightarrow \mathcal{M}(\mathbb{R})$ is the reward model, and $\gamma \in [0, 1)$ is a discount factor.[1] Let $r(x, a) = \mathbb{E}\left[\mathcal{R}(\cdot | x, a)\right]$, and assume that $r$ is uniformly bounded by $R_{\max}$. A measurable mapping $\pi : \mathcal{X} \rightarrow \mathcal{A}$ is called a *policy*. As usual, $V^\pi$ and $Q^\pi$ denote the value and action-value function for $\pi$, and $V^*$ and $Q^*$ denote the corresponding value functions for the optimal policy $\pi^*$ [5].

Our algorithm is couched in the framework of API [7]. A standard API algorithm starts with an initial policy $\pi_0$. At the $(k+1)^{\text{th}}$ iteration, given a policy $\pi_k$, the algorithm approximately evaluates $\pi_k$ to find $\hat{Q}_k$, usually as an approximate fixed point of the Bellman operator $T^{\pi_k}$: $\hat{Q}_k \approx T^{\pi_k}\hat{Q}_k$.[2] This is called the approximate *policy evaluation* step. Then, a new policy $\pi_{k+1}$ is computed, which is greedy with respect to $\hat{Q}_k$. There are several variants of API that mostly differ on how the approximate policy evaluation is performed. Most methods attempt to exploit the structures in the value function [8, 9, 10, 11], but in some problems one might have extra information about the structure of good or optimal policies as well. This is precisely our case, since we have expert demonstrations.

To develop the algorithm, we start with regularized Bellman error minimization, which is a common flavour of policy evaluation used in API. Suppose that we want to evaluate a policy $\pi$ given a batch of data $\mathcal{D}_{\text{RL}} = \{(X_i, A_i)\}_{i=1}^n$ containing $n$ examples, and that the exact Bellman operator $T^\pi$ is known. Then, the new value function $\hat{Q}$ is computed as:

$$\hat{Q} \leftarrow \operatorname*{argmin}_{Q \in \mathcal{F}^{|\mathcal{A}|}} \|Q - T^\pi Q\|_n^2 + \lambda J^2(Q), \tag{1}$$

where $\mathcal{F}^{|\mathcal{A}|}$ is the set of action-value functions, the first term is the squared Bellman error evaluated on the data,[3] $J^2(Q)$ is the regularization penalty, which can prevent overfitting when $\mathcal{F}^{|\mathcal{A}|}$ is complex, and $\lambda > 0$ is the regularization coefficient. The regularizer $J(Q)$ measures the complexity of function $Q$. Different choices of $\mathcal{F}^{|\mathcal{A}|}$ and $J$ lead to different notions of complexity, e.g., various definitions of smoothness, sparsity in a dictionary, etc. For example, $\mathcal{F}^{|\mathcal{A}|}$ could be a reproducing kernel Hilbert space (RKHS) and $J$ its corresponding norm, i.e., $J(Q) = \|Q\|_{\mathcal{H}}$.

In addition to $\mathcal{D}_{\text{RL}}$, we have a set of expert examples $\mathcal{D}_{\text{E}} = \{(X_i, \pi_E(X_i))\}_{i=1}^m$, which we would like to take into account in the optimization process. The intuition behind our algorithm is that we want to use the expert examples to "shape" the value function where they are available, while using the RL data to improve the policy everywhere else. Hence, even if we have few demonstration examples, we can still obtain good generalization everywhere due to the RL data.

To incorporate the expert examples in the algorithm one might require that at the states $X_i$ from $\mathcal{D}_{\text{E}}$, the demonstrated action $\pi_E(X_i)$ be optimal, which can be expressed as a large-margin constraint:

$Q(X_i, \pi_E(X_i)) - \max_{a \in \mathcal{A} \setminus \pi_E(X_i)} Q(X_i, a) \geq 1$. However, this might not always be feasible, or desirable (if the expert itself is not optimal), so we add slack variables $\xi_i \geq 0$ to allow occasional violations of the constraints (similar to soft vs. hard margin in the large-margin literature [12]). The policy evaluation step can then be written as the following constrained optimization problem:

$$\hat{Q} \leftarrow \operatorname*{argmin}_{Q \in \mathcal{F}^{|\mathcal{A}|}, \xi \in \mathbb{R}_+^m} \|Q - T^\pi Q\|_n^2 + \lambda J^2(Q) + \frac{\alpha}{m} \sum_{i=1}^m \xi_i \tag{2}$$

$$\text{s.t.} \quad Q(X_i, \pi_E(X_i)) - \max_{a \in \mathcal{A} \setminus \pi_E(X_i)} Q(X_i, a) \geq 1 - \xi_i. \qquad \text{for all } (X_i, \pi_E(X_i)) \in \mathcal{D}_E$$

The parameter $\alpha$ balances the influence of the data obtained by the RL algorithm (generally by trial-and-error) vs. the expert data. When $\alpha = 0$, we obtain (1), while when $\alpha \to \infty$, we essentially solve a structured classification problem based on the expert's data [13]. Note that the right side of the constraints could also be multiplied by a coefficient $\Delta_i > 0$, to set the size of the acceptable margin between the $Q(X_i, \pi_E(X_i))$ and $\max_{a \in \mathcal{A} \setminus \pi_E(X_i)} Q(X_i, a)$. Such a coefficient can then be set adaptively for different examples. However, this is beyond the scope of the paper.

The above constrained optimization problem is equivalent to the following unconstrained one:

$$\hat{Q} \leftarrow \operatorname*{argmin}_{Q \in \mathcal{F}^{|\mathcal{A}|}} \|Q - T^\pi Q\|_n^2 + \lambda J^2(Q) + \frac{\alpha}{m} \sum_{i=1}^m \left[1 - \left(Q(X_i, \pi_E(X_i)) - \max_{a \in \mathcal{A} \setminus \pi_E(X_i)} Q(X_i, a)\right)\right]_+ \tag{3}$$

where $[1 - z]_+ = \max\{0, 1 - z\}$ is the hinge loss.

In many problems, we do not have access to the exact Bellman operator $T^\pi$, but only to samples $\mathcal{D}_{\text{RL}} = \{(X_i, A_i, R_i, X_i')\}_{i=1}^n$ with $R_i \sim \mathcal{R}(\cdot|X_i, A_i)$ and $X_i' \sim P(\cdot|X_i, A_i)$. In this case, one might want to use the empirical Bellman error $\|Q - \hat{T}^\pi Q\|_n^2$ (with $(\hat{T}^\pi Q)(X_i, A_i) \triangleq R_i + \gamma Q(X_i', \pi(X_i'))$ for $1 \leq i \leq n$) instead of $\|Q - T^\pi Q\|_n^2$. It is known, however, that this is a biased estimate of the Bellman error, and does not lead to proper solutions [14]. One approach to address this issue is to use the modified Bellman error [14]. Another approach is to use Projected Bellman error, which leads to an LSTD-like algorithm [8]. Using the latter idea, we formulate our optimization as:

$$\hat{Q} \leftarrow \operatorname*{argmin}_{Q \in \mathcal{F}^{|\mathcal{A}|}, \xi \in \mathbb{R}_+^m} \left\|Q - \hat{h}_Q\right\|_n^2 + \lambda J^2(Q) + \frac{\alpha}{m} \sum_{i=1}^m \xi_i \tag{4}$$

$$\text{s.t.} \quad \hat{h}_Q = \operatorname*{argmin}_{h \in \mathcal{F}^{|\mathcal{A}|}} \left[\left\|h - \hat{T}^\pi Q\right\|_n^2 + \lambda_h J^2(h)\right]$$

$$Q(X_i, \pi_E(X_i)) - \max_{a \in \mathcal{A} \setminus \pi_E(X_i)} Q(X_i, a) \geq 1 - \xi_i. \qquad \text{for all } (X_i, \pi_E(X_i)) \in \mathcal{D}_E$$

Here $\lambda_h > 0$ is the regularization coefficient for $\hat{h}_Q$, which might be different from $\lambda$. For some choices of the function space $\mathcal{F}^{|\mathcal{A}|}$ and the regularizer $J$, the estimate $\hat{h}_Q$ can be found in closed-form. For example, one can use linear function approximators $h(\cdot) = \phi(\cdot)^\top \mathbf{u}$ and $Q(\cdot) = \phi(\cdot)^\top \mathbf{w}$ where $\mathbf{u}, \mathbf{w} \in \mathbb{R}^p$ are parameter vectors and $\phi(\cdot) \in \mathbb{R}^p$ is a vector of $p$ linearly independent basis functions defined over the space of state-action pairs. Using $L_2$-regularization, $J^2(h) = \mathbf{u}^\top \mathbf{u}$ and $J^2(Q) = \mathbf{w}^\top \mathbf{w}$, the best parameter vector $\mathbf{u}^*$ can be obtained as a function of $\mathbf{w}$ by solving a ridge regression problem:

$$\mathbf{u}^*(\mathbf{w}) = \left(\mathbf{\Phi}^\top \mathbf{\Phi} + n\lambda_h \mathbf{I}\right)^{-1} \mathbf{\Phi}^\top (\mathbf{r} + \gamma \mathbf{\Phi}' \mathbf{w}),$$

where $\mathbf{\Phi}$, $\mathbf{\Phi}'$ and $\mathbf{r}$ are the feature matrices and reward vector, respectively: $\mathbf{\Phi} = (\phi(Z_1), \ldots, \phi(Z_n))^\top$, $\mathbf{\Phi}' = (\phi(Z_1'), \ldots, \phi(Z_n'))^\top$, $\mathbf{r} = (R_1, \ldots, R_n)^\top$, with $Z_i = (X_i, A_i)$ and $Z_i' = (X_i', \pi(X_i'))$ (for data belonging to $\mathcal{D}_{\text{RL}}$). More generally, as discussed above, we might choose the function space $\mathcal{F}^{|\mathcal{A}|}$ to be a reproducing kernel Hilbert space (RKHS) and $J$ to be its corresponding norm, which provides the flexibility of working with a nonparametric representation while still having a closed-form solution for $\hat{h}_Q$. We do not provide the detail of formulation here due to space constraints.

The approach presented so far tackles the policy evaluation step of the API algorithm. As usual in API, we alternate this step with the policy improvement step (i.e., greedification). The resulting algorithm is called Approximate Policy Iteration with Demonstration (APID).

Up to this point, we have left open the problem of how the datasets $\mathcal{D}_{\mathrm{RL}}$ and $\mathcal{D}_{\mathrm{E}}$ are obtained. These datasets might be regenerated at each iteration, or they might be reused, depending on the availability of the expert and the environment. In practice, if the expert data is rare, $\mathcal{D}_{\mathrm{E}}$ will be a single fixed batch, but $\mathcal{D}_{\mathrm{RL}}$ could be increased, e.g., by running the most current policy (possibly with some exploration) to collect more data. The approach used should be tailored to the application. Note that the values of the regularization coefficients as well as $\alpha$ should ideally change from iteration to iteration as a function of the number of samples as well as the value function $Q^{\pi_k}$. The choice of these parameters might be automated by model selection [15].

## 3   Theoretical Analysis

In this section we focus on the $k^{\text{th}}$ iteration of APID and consider the solution $\hat{Q}$ to the optimization problem (2). The theoretical contribution is an upper bound on the true Bellman error of $\hat{Q}$. Such an upper bound allows us to use error propagation results [16, 17] to provide a performance guarantee on the value of the outcome policy $\pi_K$ (the policy obtained after $K$ iterations of the algorithm) compared to the optimal value function $V^*$. We make the following assumptions in our analysis.

**Assumption A1 (Sampling)** $\mathcal{D}_{\mathrm{RL}}$ contains $n$ independent and identically distributed (i.i.d.) samples $(X_i, A_i) \overset{\text{i.i.d.}}{\sim} \nu_{\mathrm{RL}} \in \mathcal{M}(\mathcal{X} \times \mathcal{A})$ where $\nu_{\mathrm{RL}}$ is a fixed distribution (possibly dependent on $k$) and the states in $\mathcal{D}_{\mathrm{E}} = \{(X_i, \pi_E(X_i))\}_{i=1}^{m}$ are also drawn i.i.d. $X_i \overset{\text{i.i.d.}}{\sim} \nu_{\mathrm{E}} \in \mathcal{M}(\mathcal{X})$ from an expert distribution $\nu_{\mathrm{E}}$. $\mathcal{D}_{\mathrm{RL}}$ and $\mathcal{D}_{\mathrm{E}}$ are independent from each other. We denote $N = n + m$.

**Assumption A2 (RKHS)** The function space $\mathcal{F}^{|\mathcal{A}|}$ is an RKHS defined by a kernel function $\mathsf{K} : (\mathcal{X} \times \mathcal{A}) \times (\mathcal{X} \times \mathcal{A}) \to \mathbb{R}$, i.e., $\mathcal{F}^{|\mathcal{A}|} = \left\{ z \mapsto \sum_{i=1}^{N} w_i \mathsf{K}(z, Z_i) : w \in \mathbb{R}^N \right\}$ with $\{Z_i\}_{i=1}^{N} = \mathcal{D}_{\mathrm{RL}} \cup \mathcal{D}_{\mathrm{E}}$. We assume that $\sup_{z \in \mathcal{X} \times \mathcal{A}} \mathsf{K}(z, z) \leq 1$. Moreover, the function space $\mathcal{F}^{|\mathcal{A}|}$ is $Q_{\max}$-bounded.

**Assumption A3 (Function Approximation Property)** For any policy $\pi$, $Q^{\pi} \in \mathcal{F}^{|\mathcal{A}|}$.

**Assumption A4 (Expansion of Smoothness)** For all $Q \in \mathcal{F}^{|\mathcal{A}|}$, there exist constants $0 \leq L_R, L_P < \infty$, depending only on the MDP and $\mathcal{F}^{|\mathcal{A}|}$, such that for any policy $\pi$, $J(T^{\pi} Q) \leq L_R + \gamma L_P J(Q)$.

**Assumption A5 (Regularizers)** The regularizer functionals $J : B(\mathcal{X}) \to \mathbb{R}$ and $J : B(\mathcal{X} \times \mathcal{A}) \to \mathbb{R}$ are pseudo-norms on $\mathcal{F}$ and $\mathcal{F}^{|\mathcal{A}|}$, respectively,[4] and for all $Q \in \mathcal{F}^{|\mathcal{A}|}$ and $a \in \mathcal{A}$, we have $J(Q(\cdot, a)) \leq J(Q)$.

Some of these assumptions are quite mild, while some are only here to simplify the analysis, but are not necessary for practical application of the algorithm. For example, the i.i.d. assumption A1 can be relaxed using *independent block technique* [18] or other techniques to handle dependent data, e.g., [19]. The method is certainly not specific to RKHS (Assumption A2), so other function spaces can be used without much change in the proof. Assumption A3 holds for "rich" enough function spaces, e.g., universal kernels satisfy it for reasonable $Q^{\pi}$. Assumption A4 ensures that if $Q \in \mathcal{F}^{|\mathcal{A}|}$ then $T^{\pi} Q \in \mathcal{F}^{|\mathcal{A}|}$. It holds if $\mathcal{F}^{|\mathcal{A}|}$ is rich enough and the MDP is "well-behaving". Assumption A5 is mild and ensures that if we control the complexity of $Q \in \mathcal{F}^{|\mathcal{A}|}$, the complexity of $Q(\cdot, a) \in \mathcal{F}$ is controlled too. Finally, note that focusing on the case when we have access to the true Bellman operator simplifies the analysis while allowing us to gain more understanding about APID. We are now ready to state the main theorem of this paper.

**Theorem 1.** *For any fixed policy $\pi$, let $\hat{Q}$ be the solution to the optimization problem (2) with the choice of $\alpha > 0$ and $\lambda > 0$. If Assumptions A1–A5 hold, for any $n, m \in \mathbb{N}$ and $0 < \delta < 1$, with probability at least $1 - \delta$ we have*

$$\left\| \hat{Q} - T^\pi \hat{Q} \right\|_{\nu_{RL}}^2 \leq 64 Q_{max} \frac{\sqrt{n+m}}{n} \left( \frac{(1 + \gamma L_P)\sqrt{R_{max}^2 + \alpha}}{\sqrt{\lambda}} + L_R \right) +$$

$$\min \left\{ 2\alpha \mathbb{E}_{X \sim \nu_E} \left[ \left[ 1 - \left( Q^\pi(X, \pi_E(X)) - \max_{a \in \mathcal{A} \setminus \pi_E(X)} Q^\pi(X, a) \right) \right]_+ \right] + \lambda J^2(Q^\pi), \right.$$

$$2 \left\| Q^{\pi_E} - T^\pi Q^{\pi_E} \right\|_{\nu_{RL}}^2 + 2\alpha \mathbb{E}_{X \sim \nu_E} \left[ \left[ 1 - \left( Q^{\pi_E}(X, \pi_E(X)) - \max_{a \in \mathcal{A} \setminus \pi_E(X)} Q^{\pi_E}(X, a) \right) \right]_+ \right] +$$

$$\left. \lambda J^2(Q^{\pi_E}) \right\} + 4Q_{max}^2 \left( \sqrt{\frac{2\ln(4/\delta)}{n}} + \frac{6\ln(4/\delta)}{n} \right) + \alpha \frac{20(1 + 2Q_{max})\ln(8/\delta)}{3m}.$$

The proof of this theorem is in the supplemental material. Let us now discuss some aspects of the result. The theorem guarantees that when the amount of RL data is large enough ($n \gg m$), we indeed minimize the Bellman error if we let $\alpha \to 0$. In that case, the upper bound would be $O_P(\frac{1}{\sqrt{n\lambda}}) + \min\{\lambda J^2(Q^\pi), 2 \|Q^{\pi_E} - T^\pi Q^{\pi_E}\|_{\nu_{RL}}^2 + \lambda J^2(Q^{\pi_E})\}$. Considering only the first term inside $\min$, the upper bound is minimized by the choice of $\lambda = [n^{1/3}J^{4/3}(Q^\pi)]^{-1}$, which leads to $O_P(J^{2/3}(Q^\pi) n^{-1/3})$ behaviour of the upper bound. The bound shows that the difficulty of learning depends on $J(Q^\pi)$, which is the complexity of the true (but unknown) action-value function $Q^\pi$ measured according to $J$ in $\mathcal{F}^{|\mathcal{A}|}$. Note that $Q^\pi$ might be "simple" with respect to some choice of function space/regularizer, but complex in another one. The choice of $\mathcal{F}^{|\mathcal{A}|}$ and $J$ reflects prior knowledge regarding the function space and complexity measure that are suitable.

When the number of samples $n$ increases, we can afford to increase the size of the function space by making $\lambda$ smaller. Since we have two terms inside $\min$, the complexity of the problem might actually depend on $2 \|Q^{\pi_E} - T^\pi Q^{\pi_E}\|_{\nu_{RL}}^2 + \lambda J^2(Q^{\pi_E})$, which is the Bellman error of $Q^{\pi_E}$ (the true action-value function of the expert) according to $\pi$ plus the complexity of $Q^{\pi_E}$ in $\mathcal{F}^{|\mathcal{A}|}$. Roughly speaking, if $\pi$ is close to $\pi_E$, the Bellman error would be small. Two remarks are in order. First, this result does not provide a proper upper bound on the Bellman error when $m$ dominates $n$. This is to be expected, because if $\pi$ is quite different from $\pi_E$ and we do not have enough samples in $\mathcal{D}_{RL}$, we cannot guarantee that the Bellman error, which is measured according to $\pi$, will be small. But, one can still provide a guarantee by choosing a large $\alpha$ and using a margin-based error bound (cf. Section 4.1 of [20]). Second, this upper bound is not optimal, as we use a simple proof technique based on controlling the supremum of the empirical process. More advanced empirical processes techniques can be used to obtain a faster error rate (cf. [12]).

## 4 Experiments

We evaluate APID on a simulated domain, as well as a real robot path-finding task. In the simulated environment, we compare APID against other benchmarks under varying availability and optimality of the expert demonstrations. In the real robot task, we evaluate the practicality of deploying APID on a live system, especially when $\mathcal{D}_{RL}$ and $\mathcal{D}_E$ are *both* expensive to obtain.

### 4.1 Car Brake Control Simulation

In the vehicle brake control simulation [21], the agent's goal is reach a target velocity, then maintain that target. It can either press the acceleration pedal or the brake pedal, but not both simultaneously. A state is represented by four continuous-valued features: target and current velocities, current positions of brake pedal and acceleration pedal. Given a state, the learned policy has to output one of five actions: *acceleration up, acceleration down, brake up, brake down, do nothing*. The reward is $-10$ times the error in velocity. The initial velocity is set to 2m/s, and the target velocity is set to 7m/s. The expert was implemented using the dynamics between the pedal pressure and output velocity, from which we calculate the optimal velocity at each state. We added random noise to the dynamics to simulate a realistic scenario, in which the output velocity is governed by factors such as friction and wind. The agent has no knowledge of the dynamics, and receives only $\mathcal{D}_E$ and $\mathcal{D}_{RL}$.

For all experiments, we used a linear Radial Basis Function (RBF) approximator for the value function and CVX, a package for specifying and solving convex programs [22], to solve the optimization

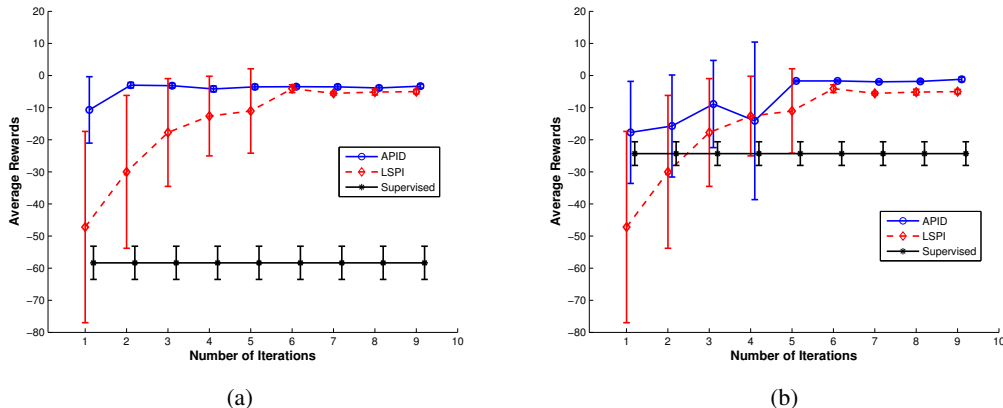

Figure 1: (a) Average reward with $m = 15$ optimal demonstrations. (b) Average reward with $m = 100$ sub-optimal demonstrations. Each iteration adds 500 new RL data to APID and LSPI, while the expert data stays the same. First iteration has $n = 500 - m$ for APID. LSPI treats all the data at this iteration RL data.

problem (4). We set $\frac{\alpha}{m}$ to 1 if the expert is optimal and 0.01 otherwise. The regularization parameter was set according to $1/\sqrt{n+m}$. We averaged results over 10 runs and computed confidence intervals as well. We compare APID with the regularized version of LSPI [6] in all the experiments. Depending on the availability of expert data, we either compare APID with the standard supervised LfD, or DAgger [1], a state-of-the-art LfD algorithm that has strong theoretical guarantees and good empirical performance when the expert is optimal. DAgger is designed to query for more demonstrations at each iteration; then, it aggregates all demonstrations and trains a new policy. The number of queries in DAgger increases linearly with the task horizon. For Dagger and supervised LfD, we use random forests to train the policy.

We first consider the case with little but optimal expert data, with task horizon 1000. At each iteration, the agent gathers more RL data using a random policy. In this case, shown in Figure 1a, LSPI performs worse than APID on average, and it also has much higher variance, especially when $\mathcal{D}_{\mathrm{RL}}$ is small. This is consistent with empirical results in [6], in which LSPI showed significant variance even for simple tasks. In the first iteration, APID has moderate variance, but it is quickly reduced in the next iteration. This is due to the fact that expert constraints impose a particular shape to the value function, as noted in Section 2. The supervised LfD performs the worst, as the amount of expert data is insufficient. Results for the case in which the agent has more but sub-optimal expert data are shown in Figure 1b. Here, with probability 0.5 the expert gives a random action rather than the optimal action. Compared to supervised LfD, APID and LSPI are both able to overcome sub-optimality in the expert's behaviour to achieve good performance, by leveraging the RL data.

Next, we consider the case of abundant demonstrations from a sub-optimal expert, who gives random actions with probability 0.25, to characterize the difference between APID and DAgger. The task horizon is reduced to 100, due to the number of demonstrations required by DAgger. As can be seen in Figure 2a, the sub-optimal demonstrations cause DAgger to diverge, because it changes the policy at each iteration, based on the newly aggregated sub-optimal demonstrations. APID, on the other hand, is able to learn a better policy by leveraging $\mathcal{D}_{RL}$. APID also outperforms LSPI (which uses the same $\mathcal{D}_{RL}$), by generalizing from $\mathcal{D}_E$ via function approximation. This result illustrates well APID's robustness to sub-optimal expert demonstrations. Figure 2b shows the result for the case of optimal and abundant demonstrations. In this case, which fits Dagger's assumptions, the performance of APID is on par with that of DAgger.

## 4.2 Real Robot Path Finding

We now evaluate the practicality of APID on a real robot path-finding task and compare it with LSPI and supervised LfD, using only one demonstrated trajectory. We do not assume that the expert is optimal (and abundant), and therefore do not include DAgger, which was shown to perform poorly for this case. In this task, the robot needs to get to the goal in an unmapped environment by learning

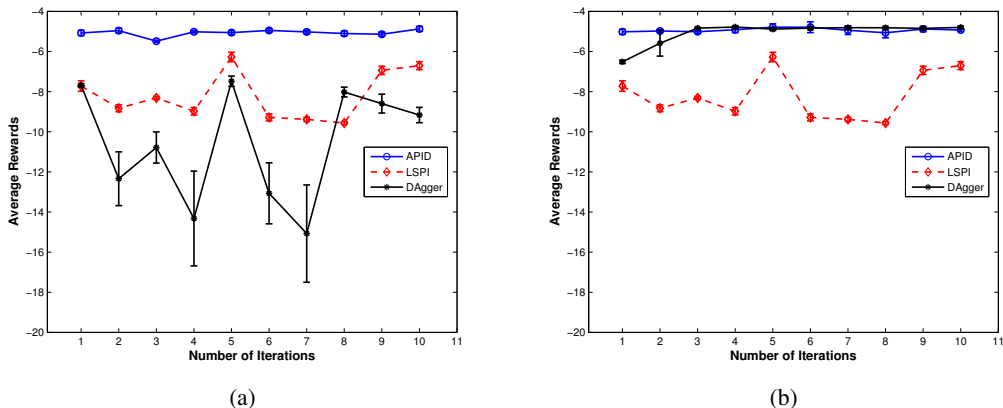

Figure 2: (a) Performance with a sub-optimal expert. (b) Performance with an optimal expert. Each iteration ($X$-axis) adds 100 new expert data points to APID and DAgger. We use $n = 3000 - m$ for APID. LSPI treats all data as RL data.

to avoid obstacles. We use an iRobot Create equipped with Kinect RGB-depth sensor and a laptop. We encode the Kinect observations with $1 \times 3$ grid cells (each 1m $\times$ 1m). The robot also has three bumpers to detect a collision from the front, left, and right. Figures 3a and 3b show a picture of the robot and its environment. In order to reach the goal, the robot needs to turn left to avoid a first box and wall on the right, while not turning too much, to avoid the couch. Next, the robot must turn right to avoid a second box, but make sure not to turn too much or too soon to avoid colliding with the wall or first box. Then, the robot needs to get into the hallway, turn right, and move forward to reach the goal position; the goal position is 6m forward and 1.5m right from the initial position.

The state space is represented by 3 non-negative integer features to represent number of point clouds produced by Kinect in each grid cell, and 2 continuous features (robot position). The robot has three discrete actions: *turn left, turn right, and move forward.* The reward is minus the distance to the goal, but if the robot's front bumper is pressed and the robot moves forward, it receives a penalty equal to 2 times the current distance to the goal. If the robot's left bumper is pressed and the robot does not turn right, and vice-versa, it also receives 2 times the current distance to the goal. The robot outputs actions at a rate of 1.7Hz.

We started from a single trajectory of demonstration, then incrementally added only RL data. The number of data points added varied at each iteration, but the average was 160 data points, which is around 1.6 minutes of exploration using $\epsilon$-greedy exploration policy (decreasing $\epsilon$ over iterations). For 11 iterations, the training time was approximately 18 minutes. Initially, $\frac{\alpha}{m}$ was set to 0.9, then it was decreased as new data was acquired. To evaluate the performance of each algorithm, we ran each iteration's policy for a task horizon of 100 ($\approx$ 1min); we repeated each iteration 5 times, to compute the mean and standard deviation.

As shown in Figure 3c, APID outperformed both LSPI and supervised LfD; in fact, these two methods could not reach the goal. The supervised LfD kept running into the couch, as state distributions of expert and robot differed, as noted in [1]. LSPI had a problem due to exploring unnecessary states; specifically, the $\epsilon$-greedy policy of LSPI explored regions of state space that were not relevant in learning the optimal plan, such as the far left areas from the initial position. $\epsilon$-greedy policy of APID, on the other hand, was able to leverage the expert data to efficiently explore the most relevant states and avoid unnecessary collisions. For example, it learned to avoid the first box in the first iteration, then explored states near the couch, where supervised LfD failed. Table 1 gives the time it took for the robot to reach the goal (within 1.5m). Only iterations 9, 10 and 11 of APID reached the goal. Note that the times achieved by APID (iteration 11) are similar to the expert.

Table 1: Average time to reach the goal

| Average Vals | Demonstration | APID-9th | APID-10th | APID-11th |
|---|---|---|---|---|
| Time To Goal(s) | 35.9 | $38.4 \pm 0.81$ | $37.7 \pm 0.84$ | $36.1 \pm 0.24$ |

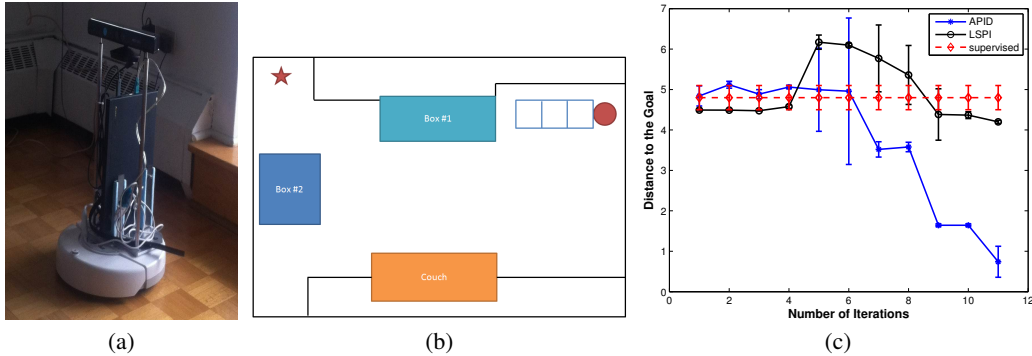

<div align="center">(a)&emsp;&emsp;&emsp;&emsp;&emsp;&emsp;&emsp;&emsp;&emsp;(b)&emsp;&emsp;&emsp;&emsp;&emsp;&emsp;&emsp;&emsp;&emsp;(c)</div>

Figure 3: (a) Picture of iRobot Create equipped with Kinect. (b) Top-down view of the environment. The star represents the goal, the circle represents the initial position, black lines indicate walls, and the three grid cells represent the vicinity of the Kinect. (c) Distance to the goal for LSPI, APID and supervised LfD with random forest.

## 5&emsp;Discussion

We proposed an algorithm that learns from limited demonstrations by leveraging a state-of-the-art API method. To our knowledge, this is the first LfD algorithm that learns from few and/or suboptimal demonstrations. Most LfD methods focus on solving the issue of violation of i.i.d. data assumptions by changing the policy slowly [23], by reducing the problem to online learning [1], by querying the expert [2] or by obtaining corrections from the expert [3]. These methods all assume that the expert is optimal or close-to-optimal, and demonstration data is abundant. The TAMER system [24] uses rewards provided by the expert (and possibly blended with MDP rewards), instead of assuming that an action choice is provided. There are a few Inverse RL methods that do not assume optimal experts [25, 26], but their focus is on learning the reward function rather than on planning. Also, these methods require a model of the system dynamics, which is typically not available in practice.

In the simulated environment, we compared our method with DAgger (a state-of-the-art LfD method) as well as with a popular API algorithm, LSPI. We considered four scenarios: very few but optimal demonstrations, a reasonable number of sub-optimal demonstrations, abundant sub-optimal demonstrations, and abundant optimal demonstrations. In the first three scenarios, which are more realistic, our method outperformed the others. In the last scenario, in which the standard LfD assumptions hold, APID performed just as well as DAgger. In the real robot path-finding task, our method again outperformed LSPI and supervised LfD. LSPI suffered from inefficient exploration, and supervised LfD was affected by the violation of the i.i.d. assumption, as pointed out in [1]. We note that APID accelerated API by utilizing demonstration data. Previous approaches [27, 28] accelerated policy search, e.g. by using LfD to find initial policy parameters. In contrast, APID leverages the expert data to shape the policy throughout the planning.

The most similar to our work, in terms of goals, is [29], where the agent is given multiple sub-optimal trajectories, and infers a hidden desired trajectory using Expectation Maximization and Kalman Filtering. However, their approach is less general, as it assumes a particular noise model in the expert data, whereas APID is able to handle demonstrations that are sub-optimal non-uniformly along the trajectory.

In future work, we will explore more applications of APID and study its behaviour with respect to $\Delta_i$. For instance, in safety-critical applications, large $\Delta_i$ could be used at critical states.

**Acknowledgements**
Funding for this work was provided by the NSERC Canadian Field Robotics Network, Discovery Grants Program, and Postdoctoral Fellowship Program, as well as by the CIHR (CanWheel team), and the FQRNT (Regroupements stratégiques INTER et REPARTI).

## Footnotes

[1]For a space $\Omega$ with $\sigma$-algebra $\sigma_\Omega$, $\mathcal{M}(\Omega)$ denotes the set of all probability measures over $\sigma_\Omega$.

[2]For discrete state spaces, $(T^{\pi_k}Q)(x, a) = r(x, a) + \gamma \sum_{x'} P(x'|x, a)Q(x', \pi_k(x'))$.

[3]$\|Q - T^\pi Q\|_n^2 \triangleq \frac{1}{n} \sum_{i=1}^n |Q(X_i, A_i) - (T^\pi Q)(X_i, A_i)|^2$ with $(X_i, A_i)$ from $\mathcal{D}_{\text{RL}}$.

[4] $B(\mathcal{X})$ and $B(\mathcal{X} \times \mathcal{A})$ denote the space of bounded measurable functions defined on $\mathcal{X}$ and $\mathcal{X} \times \mathcal{A}$. Here we are slightly abusing notation as the same symbol is used for the regularizer over both spaces. However, this should not cause any confusion since the identity of the regularizer should always be clear from the context.

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
