[Supplementary Material]

# Learning from Limited Demonstrations:
# Proof of Theorem 1

**Beomjoon Kim**
School of Computer Science
McGill University
Montreal, Quebec, Canada

**Amir-massoud Farahmand**
School of Computer Science
McGill University
Montreal, Quebec, Canada

**Joelle Pineau**
School of Computer Science
McGill University
Montreal, Quebec, Canada

**Doina Precup**
School of Computer Science
McGill University
Montreal, Quebec, Canada

Recall that we want to analyze the $k^{\text{th}}$ iteration of APID. We consider the solution $\hat{Q}$ to the following optimization problem:

$$\hat{Q} \leftarrow \underset{Q \in \mathcal{F}^{|\mathcal{A}|}, \xi \in \mathbb{R}^m_+}{\operatorname{argmin}} \|Q - T^\pi Q\|_n^2 + \lambda J^2(Q) + \frac{\alpha}{m} \sum_{i=1}^m \xi_i \tag{1}$$

$$\text{s.t.} \quad Q(X_i, \pi_E(X_i)) - \max_{a \in \mathcal{A} \setminus \pi_E(X_i)} Q(X_i, a) \geq 1 - \xi_i. \qquad \text{for all } (X_i, \pi_E(X_i)) \in \mathcal{D}_{\text{E}}$$

This optimization is equivalent to the following unconstrained one:

$$\hat{Q} \leftarrow \underset{Q \in \mathcal{F}^{|\mathcal{A}|}}{\operatorname{argmin}} \|Q - T^\pi Q\|_n^2 + \lambda J^2(Q) + \frac{\alpha}{m} \sum_{i=1}^m \left[ 1 - \left( Q(X_i, \pi_E(X_i)) - \max_{a \in \mathcal{A} \setminus \pi_E(X_i)} Q(X_i, a) \right) \right]_+ \tag{2}$$

where $[1 - z]_+ = \max\{0, 1 - z\}$ is the hinge loss.

For the convenience of the reader, we quote the assumptions and the statement of the theorem again.

**Assumption A1 (Sampling)** $\mathcal{D}_{\text{RL}}$ contains $n$ independent and identically distributed (i.i.d.) samples $(X_i, A_i) \overset{\text{i.i.d.}}{\sim} \nu_{\text{RL}} \in \mathcal{M}(\mathcal{X} \times \mathcal{A})$ where $\nu_{\text{RL}}$ is a fixed distribution (possibly dependent on $k$) and the states in $\mathcal{D}_{\text{E}} = \{(X_i, \pi_E(X_i)\}_{i=1}^m$ are also drawn i.i.d. $X_i \overset{\text{i.i.d.}}{\sim} \nu_{\text{E}} \in \mathcal{M}(\mathcal{X})$ from an expert distribution $\nu_{\text{E}}$. $\mathcal{D}_{\text{RL}}$ and $\mathcal{D}_{\text{E}}$ are independent from each other. We denote $N = n + m$.

**Assumption A2 (RKHS)** The function space $\mathcal{F}^{|\mathcal{A}|}$ is an RKHS defined by a kernel function $\mathsf{K} : (\mathcal{X} \times \mathcal{A}) \times (\mathcal{X} \times \mathcal{A}) \to \mathbb{R}$, i.e., $\mathcal{F}^{|\mathcal{A}|} = \left\{ z \mapsto \sum_{i=1}^N w_i \mathsf{K}(z, Z_i) : w \in \mathbb{R}^N \right\}$ with $\{Z_i\}_{i=1}^N = \mathcal{D}_{\text{RL}} \cup \mathcal{D}_{\text{E}}$. We assume that $\sup_{z \in \mathcal{X} \times \mathcal{A}} \mathsf{K}(z, z) \leq 1$. Moreover, the function space $\mathcal{F}^{|\mathcal{A}|}$ is $Q_{\max}$-bounded.

**Assumption A3 (Function Approximation Property)** For any policy $\pi$, $Q^\pi \in \mathcal{F}^{|\mathcal{A}|}$.

**Assumption A4 (Expansion of Smoothness)** For all $Q \in \mathcal{F}^{|\mathcal{A}|}$, there exist constants $0 \leq L_R, L_P < \infty$, depending only on the MDP and $\mathcal{F}^{|\mathcal{A}|}$, such that for any policy $\pi$, $J(T^\pi Q) \leq L_R + \gamma L_P J(Q)$.

**Assumption A5 (Regularizers)** The regularizer functionals $J : B(\mathcal{X}) \to \mathbb{R}$ and $J : B(\mathcal{X} \times \mathcal{A}) \to \mathbb{R}$ are pseudo-norms on $\mathcal{F}$ and $\mathcal{F}^{|\mathcal{A}|}$, respectively,[1] and for all $Q \in \mathcal{F}^{|\mathcal{A}|}$ and $a \in \mathcal{A}$, we have $J(Q(\cdot, a)) \leq J(Q)$.

**Theorem 1.** *For any fixed policy $\pi$, let $\hat{Q}$ be the solution to the optimization problem* (1) *with the choice of $\alpha > 0$ and $\lambda > 0$. If Assumptions A1–A5 hold, for any $n, m \in \mathbb{N}$ and $0 < \delta < 1$, with probability at least $1 - \delta$ we have*

$$\left\| \hat{Q} - T^\pi \hat{Q} \right\|_{\nu_{RL}}^2 \leq 64 Q_{max} \frac{\sqrt{n+m}}{n} \left( \frac{(1 + \gamma L_P)\sqrt{R_{max}^2 + \alpha}}{\sqrt{\lambda}} + L_R \right) +$$

$$\min \left\{ 2\alpha \mathbb{E}_{X \sim \nu_E} \left[ \left[ 1 - \left( Q^\pi(X, \pi_E(X)) - \max_{a \in \mathcal{A} \backslash \pi_E(X)} Q^\pi(X, a) \right) \right]_+ \right] + \lambda J^2(Q^\pi), \right.$$

$$\left. 2 \| Q^{\pi_E} - T^\pi Q^{\pi_E} \|_{\nu_{RL}}^2 + 2\alpha \mathbb{E}_{X \sim \nu_E} \left[ \left[ 1 - \left( Q^{\pi_E}(X, \pi_E(X)) - \max_{a \in \mathcal{A} \backslash \pi_E(X)} Q^{\pi_E}(X, a) \right) \right]_+ \right] + \lambda J^2(Q^{\pi_E}) \right\}$$

$$+ 4 Q_{max}^2 \left( \sqrt{\frac{2 \ln(4/\delta)}{n}} + \frac{6 \ln(4/\delta)}{n} \right) + \alpha \frac{20(1 + 2Q_{max}) \ln(8/\delta)}{3m}.$$

To prove this theorem, we first present a simple auxiliary result.

**Lemma 2** (Noncentrail Tail Inequality). *Let $X_1, \ldots, X_n \in \mathcal{X}$ be nonnegative i.i.d. random variables bounded by $L > 0$ almost surely. For any fixed $\delta > 0$, with probability at least $1 - \delta$, we have*

$$\left| \frac{1}{n} \sum_{i=1}^{n} X_i - \mathbb{E}[X] \right| \leq \mathbb{E}[X] + \frac{10 L \ln(2/\delta)}{3n}.$$

*Proof.* We use the Bernstein inequality (e.g., Lemma 2 of [1]) to derive this result. First note that for any $\varepsilon > 0$, the boundedness and nonnegativity of $X$ imply that $\sigma^2 \leq \mathbb{E}[X^2] \leq L\mathbb{E}[|X|] = L\mathbb{E}[X] \leq L(\mathbb{E}[X] + \varepsilon)$. Thus, for any $\varepsilon > 0$,

$$\mathbb{P} \left\{ \left| \frac{1}{n} \sum_{i=1}^{n} X_i - \mathbb{E}[X] \right| > \mathbb{E}[X] + \varepsilon \right\} \leq 2 \exp \left( -\frac{n(\mathbb{E}[X] + \varepsilon)^2}{2\sigma^2 + \frac{4L}{3}(\mathbb{E}[X] + \varepsilon)} \right)$$

$$\leq 2 \exp \left( -\frac{n(\mathbb{E}[X] + \varepsilon)^2}{(2L + \frac{4L}{3})(\mathbb{E}[X] + \varepsilon)} \right)$$

$$= 2 \exp \left( -\frac{3n(\mathbb{E}[X] + \varepsilon)}{10L} \right) \leq 2 \exp \left( -\frac{3n\varepsilon}{10L} \right),$$

where we used $\mathbb{E}[X] > 0$ in the last inequality. Rearrangement of this statement leads to the desired result. $\square$

In the proof of Theorem 1, we use the concept of Rademacher complexity (or average), so we briefly define it here [2, 3]. Let $\sigma_1, \ldots, \sigma_n$ be independent random variables with $\mathbb{P}\{\sigma_i = 1\} = \mathbb{P}\{\sigma_i = -1\} = 1/2$. For a function space $\mathcal{G} : \mathcal{X} \to \mathbb{R}$, define $R_n G = \sup_{g \in \mathcal{G}} \frac{1}{n} \sum_{i=1}^{n} \sigma_i g(X_i)$. The Rademacher complexity of $\mathcal{G}$ is $\mathbb{E}[R_n G]$, in which the expectation is w.r.t. both $\sigma$ and $X_i$. One might interpret the Rademacher complexity as a measure that quantifies the extent that a function in $\mathcal{G}$ can fit to a noise sequence of length $n$ [3].

*Proof of Theorem 1.* Fix $\delta_1 > 0$. Define the following empirical norms:

$$L_{1,n}(Q) \triangleq \| Q - T^\pi Q \|_n^2 = \frac{1}{n} \sum_{i=1}^{n} |Q(Z_i) - T^\pi Q(Z_i)|^2,$$

$$L_{2,m}(Q) \triangleq \frac{1}{m} \sum_{i=1}^{m} \left[ 1 - \frac{1}{\Delta_i} \left( Q(X_i, \pi_E(X_i)) - \max_{a \neq \pi_E(X_i)} Q(X_i, a) \right) \right]_+$$

with the understanding that in the definition of $L_{1,n}$, the random variables $Z_i = (X_i, A_i)$ are elements of $\mathcal{D}_{RL}$ and in the definition of $L_{2,m}$, the random variables $(X_i, \pi_E(X_i))$ belong to $\mathcal{D}_E$.

We also define the true norms

$$L_1(Q) \triangleq \int |Q(z) - T^\pi Q(z)|^2 \, \mathrm{d}\nu_{\mathrm{RL}}(z),$$

$$L_2(Q) \triangleq \int \left[ 1 - \frac{1}{\Delta(x)} \left( Q(x, \pi_E(x)) - \max_{a \neq \pi_E(x)} Q(x, a) \right) \right]_+ \mathrm{d}\nu_{\mathrm{E}}(x).$$

Finally define

$$L_N(Q) \triangleq L_{1,n}(Q) + \alpha L_{2,m}(Q) + \lambda J^2(Q).$$

Note that $\hat{Q} \leftarrow \mathrm{argmin}_{Q \in \mathcal{F}^{|\mathcal{A}|}} L_N(Q)$ is the solution of the optimization problem (2).[2]

Because of the optimizer property of $\hat{Q}$, we have

$$L_N(\hat{Q}) \leq$$

$$\|Q^\pi - T^\pi Q^\pi\|_n^2 + \frac{\alpha}{m} \sum_{i=1}^m \left[ 1 - \frac{1}{\Delta_i} \left( Q^\pi(X_i, \pi_E(X_i)) - \max_{a \neq \pi_E(X_i)} Q^\pi(X_i, a) \right) \right]_+ + \lambda J^2(Q^\pi) =$$

$$\frac{\alpha}{m} \sum_{i=1}^m \left[ 1 - \frac{1}{\Delta_i} \left( Q^\pi(X_i, \pi_E(X_i)) - \max_{a \neq \pi_E(X_i)} Q^\pi(X_i, a) \right) \right]_+ + \lambda J^2(Q^\pi). \tag{3}$$

We also have

$$L_N(\hat{Q}) \leq \|Q^{\pi_E} - T^\pi Q^{\pi_E}\|_n^2 +$$

$$\frac{\alpha}{m} \sum_{i=1}^m \left[ 1 - \frac{1}{\Delta_i} \left( Q^{\pi_E}(X_i, \pi_E(X_i)) - \max_{a \neq \pi_E(X_i)} Q^{\pi_E}(X_i, a) \right) \right]_+ + \lambda J^2(Q^{\pi_E}). \tag{4}$$

Moreover, we have

$$\lambda J^2(\hat{Q}) \leq L_N(\hat{Q}) \leq L_N(0) = \|0 - T^\pi 0\|_n^2 + \frac{\alpha}{m} \sum_{i=1}^m \left[ 1 - \frac{1}{\Delta_i} (0 - 0) \right]_+ + \lambda J^2(0)$$

$$\leq R_{\max}^2 + \frac{\alpha}{m} m + 0.$$

Therefore,

$$J^2(\hat{Q}) \leq \frac{R_{\max}^2 + \alpha}{\lambda}. \tag{5}$$

For $B > 0$, let us define the function space with $B$-bounded value of regularizer as $\mathcal{F}^{|\mathcal{A}|}(B) \triangleq \left\{ Q : Q \in \mathcal{F}^{|\mathcal{A}|}, J(Q) \leq B \right\}$, as well as $\mathcal{F}_\lambda^{|\mathcal{A}|} = \mathcal{F}^{|\mathcal{A}|} \left( \sqrt{\frac{R_{\max}^2 + \alpha}{\lambda}} \right)$. From (5), it is clear that $\hat{Q}$ belongs to $\mathcal{F}_\lambda^{|\mathcal{A}|}$. Now we have

$$L_1(\hat{Q}) = L_{1,n}(\hat{Q}) - L_{1,n}(\hat{Q}) + L_1(\hat{Q}) \leq L_{1,n}(\hat{Q}) + \sup_{Q \in \mathcal{F}_\lambda^{|\mathcal{A}|}} |L_{1,n}(Q) - L_1(Q)|$$

$$\leq L_N(\hat{Q}) + \sup_{Q \in \mathcal{F}_\lambda^{|\mathcal{A}|}} |L_{1,n}(Q) - L_1(Q)| \tag{6}$$

We use Rademacher complexity to control the supremum of the empirical process $\sup_{Q \in \mathcal{F}_\lambda^{|\mathcal{A}|}} |L_{1,n}(Q) - L_1(Q)|$. Since $|Q - T^\pi Q|^2$ is $(2Q_{\max})^2$-bounded and $\mathrm{Var}\left[ |Q - T^\pi Q|^2 \right] \leq$

$\mathbb{E}\left[|Q - T^\pi Q|^4\right] \le (2Q_{\max})^4$, Theorem 2.1 of Bartlett et al. [2] indicates that with probability at least $1 - \delta_1$, we have

$$\sup_{Q \in \mathcal{F}_\lambda^{|\mathcal{A}|}} |L_{1,n}(Q) - L_1(Q)| \le 4\mathbb{E}\left[R_n \mathcal{G}_\lambda\right] + \sqrt{\frac{2(2Q_{\max})^4 \ln(1/\delta_1)}{n}} + \frac{8}{3}(2Q_{\max})^2 \frac{\ln(1/\delta_1)}{n}, \quad (7)$$

where $\mathcal{G}_\lambda \triangleq \left\{ |Q - T^\pi Q|^2 \,:\, Q \in \mathcal{F}_\lambda^{|\mathcal{A}|} \right\}$.

We upper bound $\mathbb{E}\left[R_n \mathcal{G}_\lambda\right]$. We use the contraction property of the Rademacher complexity as well as the simple inequality $R_n(\mathcal{G}_1 + \mathcal{G}_2) \le R_n(\mathcal{G}_1) + R_n(\mathcal{G}_2)$ (cf. Theorem 12 of Bartlett and Mendelson [3] for both). As $\left||Q_1 - T^\pi Q_1|^2 - |Q_2 - T^\pi Q_2|^2\right| \le (4Q_{\max}) |(Q_1 - T^\pi Q_1) - (Q_2 - T^\pi Q_2)|$, the Lipschitz constant needed in the contraction property is $4Q_{\max}$. Thus,

$$\mathbb{E}\left[R_n \mathcal{G}_\lambda\right] \le 2(4Q_{\max})\mathbb{E}\left[R_n \left\{ Q - T^\pi Q : Q \in \mathcal{F}_\lambda^{|\mathcal{A}|} \right\}\right]$$

$$\le 8Q_{\max}\left(\mathbb{E}\left[R_n \mathcal{F}_\lambda^{|\mathcal{A}|}\right] + \mathbb{E}\left[R_n \left\{ T^\pi Q \,:\, Q \in \mathcal{F}_\lambda^{|\mathcal{A}|} \right\}\right]\right)$$

$$\le 8Q_{\max}\left(\mathbb{E}\left[R_n \mathcal{F}_\lambda^{|\mathcal{A}|}\right] + \mathbb{E}\left[R_n \left\{ Q \,:\, Q \in \mathcal{F}^{|\mathcal{A}|}, J(Q) \le L_R + \gamma L_P \sqrt{\frac{R_{\max}^2 + \alpha}{\lambda}} \right\}\right]\right)$$

$$= 8Q_{\max}\left(\mathbb{E}\left[R_n \mathcal{F}^{|\mathcal{A}|}\left(\sqrt{\frac{R_{\max}^2 + \alpha}{\lambda}}\right)\right] + \mathbb{E}\left[R_n \mathcal{F}^{|\mathcal{A}|}\left(L_R + \gamma L_P \sqrt{\frac{R_{\max}^2 + \alpha}{\lambda}}\right)\right]\right)$$

The behaviour of these Rademacher complexities depend on the choice of the function space and the effect of $B$ in $\mathcal{F}^{|\mathcal{A}|}(B)$ on its complexity. In the case of RKHS with data points $\{Z_i\}_{i=1}^n = \mathcal{D}_{\mathrm{RL}} \cup \mathcal{D}_{\mathrm{E}}$, we have $\mathcal{F}^{|\mathcal{A}|}(B) = \left\{ z \mapsto \sum_{i=1}^N w_i \mathsf{K}(z, Z_i) \,:\, \sum_{i,j} w_i w_j \mathsf{K}(Z_i, Z_j) \le B^2 \right\}$. In this case, it is known (cf. Lemma 22 of Bartlett and Mendelson [3]) that $R_n \mathcal{F}^{|\mathcal{A}|}(B) \le \frac{2B}{n}\sqrt{\sum_{i=1}^N \mathsf{K}(Z_i, Z_i)} \le \frac{2B\sqrt{N}}{n}$. Thus,

$$\mathbb{E}\left[R_n \mathcal{G}_\lambda\right] \le 16 Q_{\max} \frac{\sqrt{N}}{n}\left[\frac{(1 + \gamma L_P)\sqrt{R_{\max}^2 + \alpha}}{\sqrt{\lambda}} + L_R\right]. \quad (8)$$

By collecting (3), (4), (6), (7), and (8), we get that with probability at least $1 - \delta_1$,

$$L_1(\hat{Q}) = \left\|\hat{Q} - T^\pi \hat{Q}\right\|_{\nu_{\mathrm{RL}}}^2 \le$$

$$\min\left\{\frac{\alpha}{m}\sum_{i=1}^m \left[1 - \frac{1}{\Delta_i}\left(Q^\pi(X_i, \pi_E(X_i)) - \max_{a \ne \pi_E(X_i)} Q^\pi(X_i, a)\right)\right]_+ + \lambda J^2(Q^\pi),\right.$$

$$\|Q^{\pi_E} - T^\pi Q^{\pi_E}\|_n^2 + \frac{\alpha}{m}\sum_{i=1}^m \left[1 - \frac{1}{\Delta_i}\left(Q^{\pi_E}(X_i, \pi_E(X_i)) - \max_{a \ne \pi_E(X_i)} Q^{\pi_E}(X_i, a)\right)\right]_+ +$$

$$\left.\lambda J^2(Q^{\pi_E})\right\} + 64 Q_{\max} \frac{\sqrt{N}}{n}\left(\frac{(1 + \gamma L_P)\sqrt{R_{\max}^2 + \alpha}}{\sqrt{\lambda}} + L_R\right) +$$

$$(2Q_{\max})^2\left[\sqrt{\frac{2\ln(1/\delta_1)}{n}} + \frac{8}{3}\frac{\ln(1/\delta_1)}{n}\right].$$

We evoke Lemma 2 to upper bound each of three empirical terms in the right-hand side by their expectation. When we use that lemma, we set the parameter of the probability of failure equal to $\delta/4$. We also set $\delta_1 = \delta/4$. To simplify the expression, we only consider the case that $\Delta_i = 1$. After

some simplifications, we get

$$\left\| \hat{Q} - T^{\pi}\hat{Q} \right\|_{\nu_{\mathrm{RL}}}^2 \leq \min \left\{ 2\alpha\mathbb{E}_{X\sim\nu_{\mathrm{E}}} \left[ \left[ 1 - \left( Q^{\pi}(X, \pi_E(X)) - \max_{a\neq\pi_E(X)} Q^{\pi}(X, a)) \right) \right]_+ \right] + \lambda J^2(Q^{\pi}), \right.$$

$$2\left\| Q^{\pi_E} - T^{\pi}Q^{\pi_E} \right\|_{\nu_{\mathrm{RL}}}^2 + 2\alpha\mathbb{E}_{X\sim\nu_{\mathrm{E}}} \left[ \left[ 1 - \left( Q^{\pi_E}(X, \pi_E(X)) - \max_{a\neq\pi_E(X)} Q^{\pi_E}(X, a)) \right) \right]_+ \right] +$$

$$\left. \lambda J^2(Q^{\pi_E}) \right\} +$$

$$64Q_{\max}\frac{\sqrt{N}}{n} \left( \frac{(1+\gamma L_P)\sqrt{R_{\max}^2 + \alpha}}{\sqrt{\lambda}} + L_R \right) + 4Q_{\max}^2 \left( \sqrt{\frac{2\ln(4/\delta)}{n}} + \frac{6\ln(4/\delta)}{n} \right) +$$

$$\alpha\frac{20(1 + 2Q_{\max})\ln(8/\delta)}{3m},$$

with probability at least $1 - \delta$. $\qquad\qquad\square$

## Footnotes

[1] $B(\mathcal{X})$ and $B(\mathcal{X} \times \mathcal{A})$ denote the space of bounded measurable functions defined on $\mathcal{X}$ and $\mathcal{X} \times \mathcal{A}$. Here we are slightly abusing notation as the same symbol is used for the regularizer over both spaces. However, this should not cause any confusion since the identity of the regularizer should always be clear from the context.

[2]In (2), we set $\Delta_i = 1$. The loss function analyzed in the proof uses the generalized version of the optimization where $\Delta_i > 0$ might not be equal to one.