[Reviews · NeurIPS 2013]

Submitted by Assigned_Reviewer_5

This paper is published in the context of making Learning from Demonstration more robust when a limited number of demonstrations are available. Many of the low level trajectory learning LfD approaches suffer from fragile policies. This paper proposes to use Reinforcement learning to overcome this limitation. This paper falls squarely in the LfD field and does not tackle Inverse reinforcement learning, i.e. the reward function is assumed to be known to the agent rather than inferred by demonstration.

One work with a very similar flavor is that of Smart, W. and Kaelbling, L.P. "Effective Reinfrocement Learning for Mobile Robots" ICRA 2002. While algorithmically different, this work begins from a similar premise: that demonstration from a human user can be used as information to bootstrap an RL learning system. This paper leverages algorithmic advances since 2002 and leverages current advances in policy iteration algorithms. It modifies these approaches by creating a set of samples that are gained from the expert user. These samples are then input into the algorithm in a weighted fashion such that a parameter can tune the impact of this data.

In general, this paper addresses a common problem with many LfD algorithms. While it requires that the reward function be defined and available for the agent, it presents a framework that allows expert data to be integrated into a policy iteration approach. Policy iteration approaches have shown great promise in recent years for scaling RL to larger domains. One of the difficulties of the policy iteration algorithms (such as LSPI) is their dependence on a set of samples that must be collected before the agent can begin learning. However, this is not a problem when performing LfD algorithms because a set of samples is provided by the human demonstrator. Thus the coupling of these two approaches seems quite natural and well suited. This paper shows promise and is evaluated on a real robot domain. Along with follow up work it may lead to a robust useable LfD system for even more complex robotic tasks.



Detailed questions and comments:

The simulated experimental results could be enhanced by showing how the results vary with the amount of expert data provided.

Was there a D_rl in the robot experiments or were the only samples from the expert demonstration? If the expert samples were the only samples perhaps the paper could address how other samples might be gained for robot tasks especially tasks that are more complex than robot navigation. In terms of future work, it seems that demonstrating how samples from a different task (i.e. with a different reward function) can be used would provide a powerful argument for this method.

One thing that could make this paper more readable would be an algorithm box. However given the space constraints this could be quite difficult to add in.

One piece of work that addresses the limitation of learning from imperfect demonstrations:

- Grollman, D. and Billard, A. Donut as I do: Learning from failed demonstrations ICRA 2011

Summary: This paper describes a policy iteration approach to the Learning from Demonstration problem in order to create a robust policy that incorporates expert knowledge. This approach shows promise and may lead to robust useable LfD systems for more complex robotics tasks.

Submitted by Assigned_Reviewer_6

This is a well-written paper combining learning from demonstration with reinforcement learning to obtain controllers that can outperform non-optimal expert demonstration much more quickly than pure reinforcement learning would enable.

The paper provides a good balance of theory, application, and discussion.

The main theory is built on relatively mild assumptions for reinforcement learning settings and provides PAC-style bounds on the Bellman error.

I find the experiments to be compelling. They are of reasonably large size and compare against the recent state-of-the-art DAgger algorithm and a standard RL algorithm, showing improvements over both in different settings. It would be nice to see some discussion on how closely the performance in practice matches the theoretical guarantees.

Despite my enthusiasm for the paper, I strongly suggest changing the title to be more reflective of the combination of LfD and RL presented in the paper. The current title does not distinguish this work greatly from previous work on LfD/inverse optimal control that can deal with small amounts of sub-optimal demonstration.
Summary: This is a well-written paper combining learning from demonstration with reinforcement learning to obtain controllers that can outperform non-optimal expert demonstration much more quickly than pure reinforcement learning would enable. It provides a good balance of theory, application, and discussion.

Submitted by Assigned_Reviewer_7

Although the author response has addressed my questions or clarified my understanding,
I am leaving the original review intact below for reference and posterity. In addition, I will
note two things in response to the response:

- With respect to TAMER and similar reward-based methods, I believe you could convert
demonstrations to oracle-like reward. Perhaps that is a thought for later.

- In the case of adversarial demonstrations, I was really thinking of the case where they
are adversarial but you do not know it (or if you like, imagine that they are very, very
poor quality demonstrations) so that alpha isn't 0.


---original---
The paper describes an extension to the modified BRM algorithm
proposed by Antos et al. by incorporating human demonstrations as
constraints. The idea is elegant and overall the paper shows that it
is a promising area of research. I do have some questions/concerns:


Did the authors consider using the modified BRM instead of the LSTD
method? Work by Farahmand et al. (Reference 5) states that when the
linear parameterized functions are used, the solution computed by the
modified BRM is *equivalent* to LSPI; however, they go on to state that
this proposition is not true when regularized versions are used for
LSTD and BRM. It is potentially possible that you get different
results when using modified BRM.

It seems that the difference between APID and LSPI is that the former
uses some form of regularization as well as a constraint on human
demonstrations, else they would provide the same answer (as shown in
references 5 and 13). It is not clear if a comparison to LSPI is not
entirely meaningful here because essentially more domain knowledge is
being fed into APID. Along those lines, in the experiments, did the
authors use a regularized of LSPI for the comparison?

Perhaps a comparison that would strengthen the paper is the inclusion
of another established LfD method: TAMER by Knox and Stone. TAMER and
some of its variants have been known to perform well in conditions
with limited human input and work under similar conditions (using
human information as well as direct experience). A comparison to this
algorithm would help to position the work well in the space of RL and
LfD.

What aspect of the algorithm particularly takes into account the
limited number of demonstrations? There does not appear to a specific
aspect of the algorithm that maximizes the information from the humans
(in fact, any information loss from the humans is made up using direct
interaction in the world). It is simply using information from experts
as constraints.

What kind of optimization do you use for the modified BRM algorithm?
In equation 4, hQ can be obtained in closed form, but how about Q^
(how is w optimized?). How do you set the optimization and
regularization parameters?

How close to optimal should the humans be? Can you comment on the
performance if the human was adversarial? In other words, to what
extent can your assumptions on optimality of human traces be relaxed?
How do you estimate the kind of expert from whom you are getting
demonstrations?

The intro is clear and the approach is explained well. The authors
have also highlighted aspects of the approach in their experiments.
By contrast, the theoretical aspects of the work are not as
self-contained. I believe the paper would be improved by adding
further relevant details for that part of the paper (obviously, space
is at a premium in general).
Summary: The paper describes an extension to the modified BRM algorithm
proposed by Antos et al. by incorporating human demonstrations as
constraints. The idea is elegant and overall the paper shows that it
is a promising area of research.
Author Feedback

Author rebuttal: We thank the reviewers for their useful comments and suggestions. Below we answer a few of the questions that were raised.

REVIEWER #1:
Question 1: We do in fact have the simulated results with varying amounts of expert data. In Figure 2 (RIGHT), we show that if APID uses same amount of expert data, then it just does as well as DAgger.

Question 2: For robot experiments, both D_e (expert data) and D_rl were used. In the third paragraph of section 4.2, we accidentaly called D_rl “trial-and-error data”, which might have caused the confusion. We will fix this. Regarding how to gain “other samples than expert data” for complex tasks: the expert data can be sub-optimal - hence this could be data coming from a policy of a similar but different task (i.e. transfer learning). This is related to the reviewer’s future work suggestion, which we are currently exploring.

We will include an algorithm box, space permitting. We will also aim to cite the works suggested by the reviewer.

REVIEWER #2:
We greatly appreciate the reviewer’s enthusiasm towards our paper. We will try to discuss the relationship between the theoretical and empirical results, space permitting.

REVIEWER #3:
Our work is in fact using learning from demonstration with regularized LSPI, not modified BRM.

Question 1: The reviewer is correct in that modified BRM is not the same as regularized LSPI. We have in fact tried un-regularized LSPI with linear function space in our experiments, which can be shown to be the same as modified BRM with linear function space (Antos et al.). The results of the un-regularized algorithm were not as good as for the regularized one (which are reported in the paper).

Question 2: In all our experiments, we used regularized LSPI. We’ll make this clear. The comparison between regularized LSPI and APID is meaningful primarily because even though they are using the same data (D_e and D_rl), APID can leverage D_e in a direct way, while regularized LSPI simply treats it as any other data, and cannot make special use of it.

Question 3: We thank the reviewer for the suggestion of comparison to TAMER. We will consider this for the longer journal version of our paper. Note however that TAMER aims to include oracle (human) feedback in the form of rewards, whereas our approach uses demonstration trajectories. So it’s not clear they apply to the same setting, and how to ensure that the information provided to both is equivalent.

Question 4: Regarding how we use a limited number of demonstrations: available demonstrations (however few), are used to “shape” the Q function. This shaping generalizes to other states using function approximation, hence accelerating the overall learning.

Question 5: As the reviewer noted, we use the closed form solution for hQ, and then substitute it in the regularized LSTD algorithm. The resulting optimization is convex (quadratic objective with linear constraints) and we used the cvx package by Boyd et al to solve it. We set the regularization parameter to be 1/\sqrt{n+m}, where n and m correspond to sizes of expert and RL data, respectively. The alpha values are listed in the experiments. We’ll give more detail about how to set the parameters in the final version.

Question 6: The demonstrations can be completely adversarial, in which case you would set the alpha to be zero. In this sense, the optimality assumption does not have to hold.

We’ll improve the presentation of the theoretical section with the aim of making it more accessible.